# Adaptive NN Consensus Control for Second-Order Nonlinear Multi-Agent Systems Against Sparse Sensor Attacks

1st Xiao Tang
*School of Electrical Engineering*
*Liaoning University of Technology*
Jinzhou, China
xiu_tangx@163.com

2rd Yang Yu
*School of Electrical Engineering*
*Liaoning University of Technology*
Jinzhou, China
am_yuyang@163.com

3nd Wei Wang
*School of Electrical Engineering*
*Liaoning University of Technology*
Jinzhou, China
lgwangw@gmail.com

*Abstract*—This paper investigates the secure consensus tracking control problem of second-order nonlinear multi-agent systems against sparse sensor attacks. A secure data selector is designed to extract unattacked output data from a set of output measurements affected by sensor attacks. Subsequently, utilizing the unattacked output data, a neural network (NN) secure state observer is constructed to estimate the unavailable system states. Then, an adaptive NN consensus controller is proposed via dynamic surface control technique. The designed control method ensures that all signals of the closed-loop systems are ultimately bounded, and consensus tracking control is achieved with bounded tracking errors in the presence of sparse sensor attacks. Finally, the effectiveness of the proposed control scheme is validated through a simulation of unmanned aerial vehicle attitude control systems.

*Index Terms*—nonlinear multi-agent systems, sparse sensor attacks, secure state observer, adaptive NN control

## I. INTRODUCTION

Multi-agent systems (MASs) have gained significant attention in recent years and are widely applied in fields such as transportation systems, industrial manufacturing, and power system management [1]-[3]. However, since agents rely on communication network to transmit real-time monitoring data, and they are vulnerable to potential cyber-attacks from various signals, MASs are facing significant security challenges. For instance, in power grid systems, adversaries can disrupt grid operations through attacks like injecting false data or channel interference [4]. Therefore, studying the consensus control of MASs under cyber-attacks is crucial.

Typical cyber-attacks can be divided into two cases. The first case is denial-of-service(DoS) attacks and the second case is deception attacks, sensor attacks are one of the most common deception attacks. DoS attacks can prevent information transmission, while sensor attacks can easily modify the system data, both of which can greatly influence system stability [5]-[7]. Therefore, many efforts have been focused on DoS

and sensor attacks recently. In [8], a distributed cooperative control problem was addressed for a class of linear MASs under two types of attacks. The author in [9] developed a distributed secure consensus control with event-triggering for linear leader-following MASs under DoS attacks. The article [10] proposed an event-triggered scheme to reach consensus of systems with DoS attacks. In [11], stability analysis of stochastic systems with DoS attacks was investigated by designing an observer-based event-triggered protocol. In [12], observers were designed to estimate the states of encountered sensor attacks. In the article [13], a new dynamic event-triggered strategy was introduced that could adjust the variables online, and a resilient control strategy was further devised for the paralyzed MASs under the sensor attacks [14]. In addition, the article [15] introduced two induced parameters and adaptive laws to compensate them to avoid the effect of corrupted data. However, most of the above studies are for known nonlinear systems, and there is still little research for unknown nonlinear MASs against sparse sensor attacks.

Motivated by the above discussions, this paper solves the secure consensus tracking control problem of second-order nonlinear MASs against sparse sensor attacks. A secure data selector is proposed to extract the unattacked output data from a set of output measurements affected by sensor attacks, and an NN secure state observer is constructed to estimate the unavailable system states by using the unattacked output data. Then, we design an adaptive NN consensus controller by using the dynamic surface control technique, which ensures that all signals of the closed-loop systems are ultimately bounded, and consensus tracking is achieved with bounded tracking errors in the presence of sparse sensor attacks.

## II. PROBLEM FORMULATION

### A. Graph Theory

The commmunication topology of $N$ agents is described by the directed graph $\mathcal{G} = (\mathcal{V}, \mathcal{E})$, where the node set $\mathcal{V} = \{1, ..., N\}$ with "1, ..., N" standing for the follows. $\mathcal{E} = \{(j, i) : i, j \in \mathcal{V}, i \neq j\}$ is a set of edges, where the ordered edge $(j, i)$ implies that node $i$ can receive the

This work was supported in part by the National Natural Science Foundation of China under Grant 62273170, and in part by the National Natural Science Foundation of Liaoning Province under Grant 2023JH2/101300187, Grant JYTZD2023082, and Grant 2023-MS-300.

information from node $j$. Further, define the neighbor set of node $i$ as $\mathcal{N}_i = \{j \in \mathcal{V} | (j,i) \in \mathcal{E}\}$. Define an adjacency matrix as $\mathcal{A} = [a_{i,j}] \in R^{N \times N}$, where the weight $a_{i,j} > 0$ if and only if $j \in \mathcal{N}_i$ and $a_{i,j} = 0$ otherwise. The in-degree diagonal matrix is expressed as $\mathcal{D} = diag\{d_1, ..., d_N\}$ with $d_i = \sum_{j \in \mathcal{N}_i} a_{i,j}$. The Laplacian matrix of graph $G$ is represented as $\mathcal{L} = \mathcal{D} - \mathcal{A}$. The augmented graph is described as $\bar{\mathcal{G}} = (\bar{\mathcal{V}}, \bar{\mathcal{E}})$ with $\bar{\mathcal{V}} = \{0, 1, ..., N\}$ and $\bar{\mathcal{E}} \subseteq \bar{\mathcal{V}} \times \bar{\mathcal{V}}$, where "0" is the leader. Similarly, define $\mathcal{C} = diag\{a_{1,0}, ..., a_{N,0}\}$, and $a_{1,0} > 0$ if and only if the node $i$ can receive the information from node 0 and $a_{1,0} = 0$ otherwise, denotes the communication weight from the leader to followers, $\mathcal{B} = diag[b_1, ..., b_N]^T$ where $b_i > 0$ if the leader $0 \in \mathcal{N}_i$ and $b_i = 0$ otherwise. Then, suppose that the directed communication graph $\mathcal{G}$ has a spanning tree, $rank(\mathcal{L} + \mathcal{B}) = N$ from $(\mathcal{L} + \mathcal{B})1_N = b$ where $b = [b_1, ..., b_N]^T$ and $1_N$ is an $N$-vector of all ones. Therefore, $(\mathcal{L} + \mathcal{B})$ is invertible.

### B. Problem statement

In this paper, the framework of second-ordor MASs is described as follows. Suppose that there exist $N$ followers, labeled as agaents 1 to $N$, the $i$ th agent is described as a class of strict-feedback nonlinear systems with sensor attacks:

$$
\begin{aligned}
\dot{x}_{i,1} &= x_{i,2} + f_{i,1}(x_{i,1}) \\
\dot{x}_{i,2} &= u_i + f_{i,2}(\bar{x}_i) \\
y_{\zeta,i} &= C_i \bar{x}_i + \zeta_i(t)
\end{aligned}
\tag{1}
$$

where $i = 1, 2, ..., N$, $\bar{x}_i = [x_{i,1}, x_{i,2}]^T \in R^2$ is the unavailable state vector, $f_{i,1}(x_{i,1})$ and $f_{i,2}(\bar{x}_i)$ are the unknown continuous nonlinear functions, $u_i$ is the controller to be designed. $y_{\zeta,i} \in R^{p_i}$ is the attacked output vector with $p_i$ being the number of the output sensors. $\zeta_i(t) = [\zeta_{i,1}(t), ..., \zeta_{i,pi}(t)]^T$ denotes the injected data caused by the adversarial attacker, i.e., if the sensor $k$ ($k \in \{1, 2, ..., p_i\}$) is compromised, then $\zeta_{i,k}(t) \neq 0$ for some $t \geq 0$; otherwise $\zeta_{i,k}(t) = 0$ for all $t \geq 0$. The output distribution matrix is defined as $C_i = [E_1, ..., E_1]^T \in R^{p_i \times 2}$, $E_1 \in R^2$ is a vector with the first th element being one and the rest being zero. The unattacked output vector of the $i$th agent is $\bar{y}_i = C_i \bar{x}_i = [x_{i,1}, ..., x_{i,1}]^T \in R^{p_i}$.

The control objective of this paper is to design an adaptive NN consensus controller $u_i$ for $N$ followers (1), such that under the sparse sensor attacks, the unattacked outputs of all followers $\breve{y}_i$ converge to a small neighborhood of the virtual leader's output $y_0$.

## III. Main Result

### A. Secure Data Selector

Consider that the attack vector $\zeta_i(t)$ is $s_i$-sparse, which means that at most $s_i$ ($s_i < p_i$) elements of the vector $\zeta_i(t)$ are simultaneously nonzero. To design an effective data selector, the following assumption is introduced.

**Assumption 1**: For each agent in MASs (1), the number of simultaneously attacked sensors $s_i$ satisfies $2s_i < p_i$.

Based on Assumption 1, a median value operator $Med[\cdot]$ is designed to pick out the sensor data that is not attacked. Given

a vector $y_{\zeta,i}(t)$, rearrange its elements in increasing numerical order to produce a new vector $\hbar_i(t) = [\hbar_{i,1}(t), ..., \hbar_{i,p_i}(t)]^T$. Then, the operator $Med[\cdot]$ is designed as

$$
Med[\cdot] = \begin{cases} \hbar_{i,M_i} & if \ p_i \ is \ odd \\ \frac{1}{2}(\hbar_{i,M_i} + \hbar_{i,M_i+1}) & if \ p_i \ is \ even \end{cases}
\tag{2}
$$

where $M_i = 0.5(p_i+1)$ if $p_i$ is an odd integer, and $M_i = 0.5p_i$ otherwise. Under Assumption 1, a secure data selector of the $i$th ($i = 1, 2$) agent is given as

$$
\breve{y}_i(t) = Med[y_{\zeta,i}(t)]
\tag{3}
$$

Proposition 1: For the system (1) under sparse sensor attacks, if Assumption 1 is satisfied, then the output of data selector (3) is equal to the unattacked output, i.e., $y_i(t) = \breve{y}_i(t), \forall t \geq 0$ with $y_i(t) = [\bar{y}_i]_{1,1}$.

### B. Secure State Observer

With the help of secure data selector and adaptive NN, a secure state observer is designed to estimate the unavailbale states. The system (1) can be rewritten as

$$
\dot{x}_i = A_{i0} x_i + F_i(\bar{x}_i) + E_2 u_i
\tag{4}
$$

where $A_{i0} = \begin{bmatrix} 0 & I_1 \\ 0 & 0 \end{bmatrix}$, $F_i(\bar{x}_i) = [f_{i,1}(x_{i,1}), f_{i,2}(\bar{x}_i)]^T$.

The NN secure state state observer is constructed as

$$
\dot{\hat{x}}_i = A_{i0} \hat{\bar{x}}_i + L_i(\breve{y} - \hat{x}_{i,1}) + \hat{F}_i(\bar{\chi}_i) + E_2 u_i
\tag{5}
$$

where $\hat{\bar{x}}_i = [\hat{x}_{i,1}, \hat{x}_{i,2}]^T$ is the observation value of $\bar{x}_i$, the observation error is expressed as $\tilde{x}_i = x_i - \hat{x}_i$, $L_i = [l_{i,1}, l_{i,2}]^T$ is the observation gains to be specified, $\bar{\chi}_i = [x_{i,1}, \hat{x}_{i,2}]^T$, $\hat{F}_i(\bar{\chi}_i) = [\hat{\theta}_{i,1}^T \phi_{i,1}(x_{i,1}), \hat{\theta}_{i,2}^T \phi_{i,2}(\hat{\bar{x}}_i)]^T$.

Accroding to (4) and (5), the dynamics of $\tilde{x}_i$ are derived as

$$
\dot{\tilde{x}}_i = A_i \tilde{x}_i + F_i(\bar{x}_i) - \hat{F}_i(\bar{\chi}_i)
\tag{6}
$$

where $A_i = A_{i0} - L_i E_1^T$.

### C. Consensus Controller Design

Based on the outputs of the secure data selector and NN secure state observer, the error surfaces are defined as

$$
\begin{aligned}
z_{i,1} &= \sum_{j=1}^{M} a_{ij}(\breve{y}_i - \breve{y}_j) + b_i(\breve{y}_i - y_0) \\
z_{i,2} &= \hat{x}_{i,2} - \bar{\alpha}_{i,2}
\end{aligned}
\tag{7}
$$

and

$$
s_{i,2} = \bar{\alpha}_{i,2} - \alpha_{i,2}
\tag{8}
$$

where $i = 1, 2, ..., N$, $\alpha_{i,2}$ and $\bar{\alpha}_{i,2}$ are the virtual control law and the filterd virtual control law, respectively.

Step 1: The derivative of $z_{i,1}$ along (1), (7) and (8), is

$$
\begin{aligned}
\dot{z}_{i,1} = k_i[&z_{i,2} + s_{i,2} + \alpha_{i,2} + f_{i,1}(x_{i,1}) \\
&+ \Gamma_{i,1}(x_{j,1})] - \sum_{j=1}^{N} a_{ij} x_{j,2} - b_i \dot{y}_0
\end{aligned}
\tag{9}
$$

where $k_i = d_i + b_i$ and $\Gamma_{i,1}(x_{j,1}) = -\frac{1}{k_i} \sum_{j=1}^{M} a_{ij} f_{j,1}(x_{j,1})$.

Use two RBF NNs to approximate unknown nonlinear function $f_{i,1}(x_{i,1})$ and $\Gamma_{j,1}(x_{j,1})$. They can be described as following forms:

$$f_{i,1}(x_{i,1}) = \theta_{i,1}^{*T}\phi_{i,1}(x_{i,1}) + \varepsilon_{i,1}$$

$$\Gamma_{j,1}(x_{j,1}) = \theta_{j,1}^{*T}\phi_{j,1}(x_{j,1}) + \varepsilon_{j,1}$$

where $\theta_{i,1}^*$, $\theta_{j,1}^*$ are the optimal NN weight vectors, $\phi_{i,1}(x_{i,1})$, $\phi_{j,1}(x_{j,1})$ are the Gaussian basis functions, and $\varepsilon_{i,1}$, $\varepsilon_{j,1}$ are the minimum approximation errors, respevtively.

To stabilize (9), the first virtual control law $\alpha_{i,2}$ for the ith follower is designed as

$$\alpha_{i,2} = \frac{1}{k_i}[-c_{i,1}z_{i,1} + d_i x_{j,2} + b_i \dot{y}_0] \\ - \hat{\theta}_{i,1}^T \phi_{i,1}(x_{i,1}) - \hat{\theta}_{j,1}^T \phi_{j,1}(x_{j,1}) \quad (10)$$

where $c_{i,1}$ is a positive parameter.

Choose the Lyapunov function $V_{i,1}$ as

$$V_{i,1} = \frac{1}{2}z_{i,1}^2 + \frac{1}{2\lambda_{i,1}}\tilde{\theta}_{i,1}^T\tilde{\theta}_{i,1} + \frac{1}{2\lambda_{j,1}}\tilde{\theta}_{j,1}^T\tilde{\theta}_{j,1} \quad (11)$$

The derivative of $V_{i,1}$ with respect to time is

$$\dot{V}_{i,1} = z_{i,1}\dot{z}_{i,1} - \frac{1}{\lambda_{i,1}}\tilde{\theta}_{i,1}^T\dot{\hat{\theta}}_{i,1} - \frac{1}{\lambda_{j,1}}\tilde{\theta}_{j,1}^T\dot{\hat{\theta}}_{j,1} \\ = -c_{i,1}z_{i,1}^2 + k_i z_{i,1}z_{i,2} + k_i z_{i,1}s_{i,2} + k_i z_{i,1}(\varepsilon_{i,1} + \varepsilon_{j,1}) \\ + \frac{1}{\lambda_{i,1}}\tilde{\theta}_{i,1}^T[\lambda_{i,1}k_i z_{i,1}\tilde{\theta}_{i,1}^T\phi_{i,1}(x_{i,1}) - \dot{\hat{\theta}}_{i,1}] \\ + \frac{1}{\lambda_{j,1}}\tilde{\theta}_{j,1}^T[\lambda_{j,1}k_i z_{i,1}\tilde{\theta}_{j,1}^T\phi_{j,1}(x_{j,1}) - \dot{\hat{\theta}}_{j,1}] \quad (12)$$

Design the adaptive law as

$$\dot{\hat{\theta}}_{i,1} = \lambda_{i,1}k_i z_{i,1}\tilde{\theta}_{i,1}^T\phi_{i,1}(x_{i,1}) - \sigma_{i,1}\hat{\theta}_{i,1} \\ \dot{\hat{\theta}}_{j,1} = \lambda_{j,1}k_i z_{i,1}\tilde{\theta}_{j,1}^T\phi_{j,1}(x_{j,1}) - \sigma_{j,1}\hat{\theta}_{j,1} \quad (13)$$

where $\sigma_{i,1}$ and $\sigma_{j,1}$ are positive parameters.

From (10)-(13), $\dot{V}_{i,1}$ follows that:

$$\dot{V}_{i,1} \leq -c_{i,1}z_{i1}^2 - \frac{\sigma_{i,1}}{2\lambda_{i,1}}\left\|\tilde{\theta}_{i,1}\right\|^2 - \frac{\sigma_{j,1}}{2\lambda_{j,1}}\left\|\tilde{\theta}_{j,1}\right\|^2 \\ + k_i z_{i,1}z_{i,2} + k_i z_{i,1}s_{i,2} + k_i z_{i,1}^2 + k_i(\frac{1}{2}\varepsilon_{i,1}^2 + \frac{1}{2}\varepsilon_{j,1}^2) \\ + \frac{\sigma_{i,1}}{2\lambda_{i,1}}\left\|\theta_{i,1}^*\right\|^2 + \frac{\sigma_{j,1}}{2\lambda_{j,1}}\left\|\theta_{j,1}^*\right\|^2 \quad (14)$$

Then, in order to obtain the filtered virtual control law $\bar{\alpha}_{i,2}$, we pass $\alpha_{i,2}$ through a first order filter with a small time constant $\tau_{i,2} > 0$.

$$\tau_{i,2}\dot{\bar{\alpha}}_{i,2} + \bar{\alpha}_{i,2} = \alpha_{i,2}, \quad \bar{\alpha}_{i,2}(0) = \alpha_{i,2}(0) \quad (15)$$

Step 2: According to (1), (8), we get

$$\dot{z}_{i,2} = \dot{\hat{x}}_{i,2} - \dot{\bar{\alpha}}_{i,2} \\ = u_i + l_{i,2}\tilde{x}_{i,1} + \hat{\theta}_{i,2}^T\phi_{i,2}(\hat{\bar{x}}_i) + \tilde{\theta}_{i,2}^T\phi_{i,2}(\hat{\bar{x}}_i) \\ - \tilde{\theta}_{i,2}^T\phi_{i,2}(\hat{\bar{x}}_i) - \dot{\bar{\alpha}}_{i,2} \quad (16)$$

Choose the controller $u_i$ and the adaptive law $\dot{\hat{\theta}}_{i,2}$ as:

$$u_i = -[l_{i,2}\tilde{x}_{i,1} + \hat{\theta}_{i,2}^T\phi_{i,2}(\hat{\bar{x}}_i) + c_{i,2}z_{i,2} + \dot{\bar{\alpha}}_{i,2}] \\ \dot{\hat{\theta}}_{i,2} = \lambda_{i,2}z_{i,2}\tilde{\theta}_{i,2}^T\phi_{i,2}(\hat{\bar{x}}_i) - \sigma_{i,2}\hat{\theta}_{i,2} \quad (17)$$

Then $\dot{z}_{i,2}$ can be rewritten as

$$\dot{z}_{i,2} = -c_{i,2}z_{i,2} + \tilde{\theta}_{i,2}^T\phi_{i,2}(\hat{\bar{x}}_i) - \tilde{\theta}_{i,2}^T\phi_{i,2}(\hat{\bar{x}}_i) \quad (18)$$

Choose the Lyapunov funtion

$$V_{i,2} = \frac{1}{2}z_{i,2}^2 + \frac{1}{2\lambda_{i,2}}\tilde{\theta}_{i,2}^T\tilde{\theta}_{i,2} + \frac{1}{2}s_{i,2}^2 \quad (19)$$

The derivative of $V_{i,2}$ with respect to time is

$$\dot{V}_{i,2} \leq -c_{i,2}z_{i,2}^2 + \frac{1}{2}z_{i,2}^2 + \frac{1}{2}\left\|\tilde{\theta}_{i,2}\right\|^2 \\ + \frac{\sigma_{i,2}}{\lambda_{i,2}}\tilde{\theta}_{i,2}^T\hat{\theta}_{i,2} - \frac{1}{\tau_{i,2}}s_{i,2}^2 - s_{i,2}\dot{\bar{\alpha}}_{i,2} \quad (20)$$

According to the following the Young's inequality

$$-s_{i,2}\dot{\bar{\alpha}}_{i,2} = s_{i,2}B_{i,2} \leq |s_{i,2}B_{i,2}| \leq \frac{s_{i,2}^2\bar{B}_{i,2}^2}{2\delta_{i,2}} + 2\delta_{i,2} \\ \frac{\sigma_{i,2}}{\lambda_{i,2}}\tilde{\theta}_{i,2}^T\hat{\theta}_{i,2} \leq -\frac{\sigma_{i,2}}{\lambda_{i,2}}\tilde{\theta}_{i,2}^T\tilde{\theta}_{i,2} + \frac{\sigma_{i,2}}{\lambda_{i,2}}\theta_{i,2}^{*T}\theta_{i,2}^* \quad (21)$$

Then, (20) follows that

$$\dot{V}_{i,2} \leq -c_{i,2}z_{i,2}^2 - \frac{1}{\tau_{i,2}}s_{i,2}^2 - \frac{\sigma_{i,2}}{\lambda_{i,2}}\left\|\tilde{\theta}_{i,2}\right\|^2 + \frac{1}{2}z_{i,2}^2 \\ + \frac{1}{2}\left\|\tilde{\theta}_{i,2}\right\|^2 + \frac{\sigma_{i,2}}{\lambda_{i,2}}\left\|\theta_{i,2}^*\right\|^2 + \frac{s_{i,2}^2\bar{B}_{i,2}^2}{2\delta_{i,2}} + 2\delta_{i,2} \quad (22)$$

### D. Stability Analysis

Theorem 3.1: For nonlinear second-order systems (1), if we consider the controller (17), virtual control law (10), and the adaptive laws (13), (17), we can get the overall control scheme has the following performance:

- Close-loop system is semi-globally uniformly asymptotically stable;
- Consensus tracking errors between followers and leader converge to a small neighborhood of the virtual leader's output $y_0$.

Proof: Choosing the total Lyapunov candidate funtion $V$ as

$$V = \sum_{i=1}^{N}[V_{i,1} + V_{i,2}] \quad (23)$$

Then, the time derivative of $V$ along (11)-(13), (21), (23) is computed as

$$\dot{V} \leq \sum_{i=1}^{N} \left[ \left( -c_{i,1}z_{i,1}^2 - \frac{\sigma_{i,1}}{2\lambda_{i,1}} \left\| \tilde{\theta}_{i,1} \right\|^2 - \frac{\sigma_{j,1}}{2\lambda_{j,1}} \left\| \tilde{\theta}_{j,1} \right\|^2 \right. \right.$$
$$+ k_i z_{i,1} z_{i,2} + k_i z_{i,1} s_{i,2} + k_i z_{i,1}^2 + k_i (\frac{1}{2}\varepsilon_{i,1}^2 + \frac{1}{2}\varepsilon_{j,1}^2)$$
$$\left. + \frac{\sigma_{i,1}}{2\lambda_{i,1}} \left\| \theta_{i,1}^* \right\|^2 + \frac{\sigma_{j,1}}{2\lambda_{j,1}} \left\| \theta_{j,1}^* \right\|^2 \right) - c_{i,2}z_{i,2}^2 - \frac{1}{\tau_{i,2}}s_{i,2}^2$$
$$- \frac{\sigma_{i,2}}{\lambda_{i,2}} \tilde{\theta}_{i,2}^T \tilde{\theta}_{i,2} + \frac{1}{2}z_{i,2}^2 + \frac{1}{2} \left\| \tilde{\theta}_{i,2} \right\|^2 + \frac{\sigma_{i,2}}{\lambda_{i,2}} \theta_{i,2}^{*T}\theta_{i,2}^*$$
$$+ \frac{s_{i,2}^2 \bar{B}_{i,2}^2}{2\delta_{i,2}} + 2\delta_{i,2} \right]$$

(24)

Using the Young's inequality, we have

$$\dot{V} \leq \sum_{i=1}^{N} \left[ \left( -(c_{i,1} - 2k_i)z_{i,1}^2 - (c_{i,2} - \frac{k_i+1}{2})z_{i,2}^2 \right. \right.$$
$$- (\frac{1}{\tau_{i,2}} - \frac{k_i}{2})s_{i,2}^2 - \frac{\sigma_{i,1}}{2\lambda_{i,1}} \left\| \tilde{\theta}_{i,1} \right\|^2 - \frac{\sigma_{i,2}}{2\lambda_{i,2}} \left\| \tilde{\theta}_{i,2} \right\|^2$$
$$- \frac{\sigma_{j,1}}{2\lambda_{j,1}} \left\| \tilde{\theta}_{j,1} \right\|^2 + \frac{k_i}{2}\varepsilon_{i,1}^2 + \frac{k_i}{2}\varepsilon_{j,1}^2 + \frac{\sigma_{i,1}}{2\lambda_{i,1}} \left\| \theta_{i,1}^* \right\|^2 \quad (25)$$
$$+ \frac{\sigma_{j,1}}{2\lambda_{j,1}} \left\| \theta_{j,1}^* \right\|^2 + \frac{1}{2} \left\| \tilde{\theta}_{i,2} \right\|^2 + \frac{\sigma_{i,2}}{\lambda_{i,2}} \left\| \theta_{i,2}^* \right\|^2$$
$$+ \frac{s_{i,2}^2 \bar{B}_{i,2}^2}{2\delta_{i,2}} + 2\delta_{i,2} \right]$$

Choosing the parameters satisfy $c_{i,1} - 2k_i > 0$, $c_{i,2} - \frac{k_i+1}{2} > 0$, $\frac{1}{\tau_{i,2}} - \frac{k_i}{2} > 0$ and from the known above $\sigma_{i,1}$, $\sigma_{j,1}$, $\lambda_{i,1}$, $\lambda_{j,1}$, $\lambda_{i,2}$, $\tau_{i,1}$, $\tau_{i,2}$, $\delta_{i,2}$ are positive parameters, we can obtain that

$$\dot{V} \leq -\eta V + D \quad (26)$$

where $\eta = \min[(c_{i,1} - 2k_i, c_{i,2} - \frac{k_i+1}{2}, \frac{1}{\tau_{i,2}} - \frac{k_i}{2}] > 0$, and $D = \sum_{i=1}^{N} \left[ \frac{k_i}{2}\varepsilon_{i,1}^2 + \frac{k_i}{2}\varepsilon_{j,1}^2 + \frac{\sigma_{i,1}}{2\lambda_{i,1}} \left\| \theta_{i,1}^* \right\|^2 + \frac{\sigma_{j,1}}{2\lambda_{j,1}} \left\| \theta_{j,1}^* \right\|^2 + \frac{1}{2} \left\| \tilde{\theta}_{i,2} \right\|^2 + \frac{\sigma_{i,2}}{\lambda_{i,2}} \left\| \theta_{i,2}^* \right\|^2 + \frac{s_{i,2}^2 \bar{B}_{i,2}^2}{2\delta_{i,2}} + 2\delta_{i,2} \right]$.

According to (27), we have $V(t) \leq e^{-\eta t}V(t_0) + (\frac{D}{\eta})[1 - e^{-\eta t}]$. Then, the signals in closed-loop systems are SGUUB. Moreover, utilizing $\frac{1}{2}\|z_1\|^2 \leq V(t)$ with $z_1 = [z_{1,1}^T, ..., z_{N,1}^T]^T$, we get $\|z_{i,1}\|^2 \leq 2V(t_0)e^{-\eta t} + (\frac{2D}{\eta})[1 - e^{-\eta t}]$. According to $z_1 = (\mathcal{L} + \mathcal{B})(y - (1_N \otimes y_0))$ where $y = [\breve{y}_1, ..., \breve{y}_N]^T$ and $\otimes$ stand for the Kronecker product. Therefore, the tracking errors can be made as small as possible by choosing appropriate design parameters.

## IV. SIMULATION RESULTS

Based on the above analysis, in this section we Desige a system which has a virtual leader and four UAVs with the $\mathcal{C} = diag\{1,0,0,0\}$, and the adjacency matrix $\mathcal{A} = [0,0,0,0; 1,0,0,0; 1,0,0,0; 1,0,0,0]$. The $i$th ($i = 1,2,3,4$) UAV model is described as

$$\dot{x}_{i,1} = x_{i,2} + f_{i,1}(x_{i,1})$$
$$\dot{x}_{i,2} = u_i + f_{i,2}(\bar{x}_i) \quad (27)$$
$$y_{\zeta,i} = C_i \bar{x}_i + \zeta_i(t)$$

where $x_{i,1} = [\rho_i, \beta_i, \psi_i]^T$ are the roll, pitch, yaw angles of UAV, $x_{i,2} = [\omega_i^\rho, \omega_i^\beta, \omega_i^\psi]^T$ are the angular velocities where $-\frac{\pi}{2} < \beta_i < \frac{\pi}{2}$. Design the coefficient matrix as $\gamma_i = diag\{I_x^{-1}, I_y^{-1}, I_z^{-1}\}$ which has $I_x = 0.0027kg \cdot m^2$, $I_y = 0.0027kg \cdot m^2$, $I_z = 0.0047kg \cdot m^2$, and $u_i = [u_i^\rho, u_i^\beta, u_i^\psi]^T$ represent the roll, pitch and yaw torques. Then the output distribution matrix is $C_i = [I_3, \mathbf{0}] \in R^{3\times 6}$, define $x_i' = [x_{i,1}^T, x_{i,2}^T]^T$ with $\bar{x}_i = [x_i', ..., x_i']^T \in R^{6\times p_i}$. Therefore the real output matrix is $\bar{y}_i = C_i \bar{x}_i = [x_{i,1}, ..., x_{i,1}]^T \in R^{3\times p_i}$. We choose the number of the output sensors as $p_1 = 5$, $p_2 = 6$, $p_3 = 7$, $p_4 = 7$. Besides, the unknown functions $f_{i,1}$ and $f_{i,2}$ are given as

$$f_{i,1} = \begin{bmatrix} \omega_i^\beta \sin(\rho_i)\tan(\beta_i) + \omega_i^\psi \cos(\rho_i)\tan(\beta_i) \\ \omega_i^\beta (\cos(\rho_i) - 1) - \omega_i^\psi \sin(\rho_i) \\ \omega_i^\beta \frac{\sin(\rho_i)}{\cos(\beta_i)} + \omega_i^\psi \left( \frac{\cos(\rho_i)}{\cos(\beta_i)} - 1 \right) \end{bmatrix} \quad (28)$$

$$f_{i,2} = \begin{bmatrix} \omega_i^\beta \omega_i^\psi \frac{I_y - I_z}{I_x} \\ \omega_i^\rho \omega_i^\psi \frac{I_z - I_x}{I_y} \\ \omega_i^\rho \omega_i^\beta \frac{I_x - I_y}{I_z} \end{bmatrix} \quad (29)$$

We choose the adaptive NN membership function as

$$\mu_{F_{i,k}^1}(\hat{x}_i) = \exp[-\frac{(\hat{x}_{i,k}-8)^2}{2}], \mu_{F_{i,k}^2}(\hat{x}_i) = \exp[-\frac{(\hat{x}_{i,k}-6)^2}{2}]$$
$$\mu_{F_{i,k}^3}(\hat{x}_i) = \exp[-\frac{(\hat{x}_{i,k}-4)^2}{2}], \mu_{F_{i,k}^4}(\hat{x}_i) = \exp[-\frac{(\hat{x}_{i,k}-2)^2}{2}]$$
$$\mu_{F_{i,k}^5}(\hat{x}_i) = \exp[-\frac{(\hat{x}_{i,k}-0)^2}{2}], \mu_{F_{i,k}^6}(\hat{x}_i) = \exp[-\frac{(\hat{x}_{i,k}+2)^2}{2}]$$
$$\mu_{F_{i,k}^7}(\hat{x}_i) = \exp[-\frac{(\hat{x}_{i,k}+4)^2}{2}], \mu_{F_{i,k}^8}(\hat{x}_i) = \exp[-\frac{(\hat{x}_{i,k}+6)^2}{2}]$$
$$\mu_{F_{i,k}^9}(\hat{x}_i) = \exp[-\frac{(\hat{x}_{i,k}+8)^2}{2}]$$

(30)

and order that

$$\phi_{i,1,l}(\hat{x}_{i,1}) = \frac{\mu_{F_{i,1}^1}(\hat{x}_{i,1})}{\sum_{l=1}^{9} \mu_{F_{i,1}^l}(\hat{x}_{i,1})}, \phi_{i,2,l}(\hat{\bar{x}}_{i,2}) = \frac{\prod_{i=1}^{2} \mu_{F_{i,2}^1}(\hat{\bar{x}}_{i,2})}{\sum_{l=1}^{9} (\prod_{i=1}^{2} \mu_{F_{i,2}^l}(\hat{\bar{x}}_{i,2}))}$$

(31)

$$\phi_{i,1}(\hat{x}_{i,1}) = [\phi_{i,1,1}(\hat{x}_{i,1}), ..., \phi_{i,1,l}(\hat{x}_{i,1})]^T$$
$$\phi_{i,2}(\hat{\bar{x}}_{i,2}) = [\phi_{i,2,1}(\hat{\bar{x}}_{i,2}), ..., \phi_{i,2,l}(\hat{\bar{x}}_{i,2})]^T \quad (32)$$

As for the injected data matrices, we design $\zeta_i(t)$ to describe them as

$$\zeta_i(t) = [\zeta_i'(t), \zeta_i'(t), \zeta_i'(t)]^T, i = 1, 2, 3, 4$$
$$\zeta_1'(t) = \begin{cases} [0, x_{1,1}^T \sin(x_{1,2}), 0, \sin(t), 0]^T, t \leq 10 \\ [\cos(t), 0, x_{1,1}^T x_{1,1}, 0, 0]^T, \quad t > 10 \end{cases}$$
$$\zeta_2'(t) = [0, x_{2,1}^T \cos(x_{2,2}), 0, \cos(t), 0, 0]^T \quad (33)$$
$$\zeta_3'(t) = [0, x_{3,1}^T \cos(x_{3,2}), 0, \cos^2(t), 0, x_{3,1}^T x_{3,1}, 0]^T$$
$$\zeta_4'(t) = [0, x_{4,1}^T \cos(x_{4,2}), 0, \sin^2(t), 0, x_{4,1}^T x_{4,1}, 0]^T$$

The attitude of the virtual leader is determined by $y_0 = [y_0^\rho, y_0^\beta, y_0^\psi]^T = [0.1\pi \sin(0.1\pi t), -0.1\pi \cos(0.1\pi t), 0]^T$, and we designed other parameters as follow $c_{i,1} = diag[2, 2, 1]$, $c_{i,2} = diag[20, 20, 50]$, $\lambda_{i,1} = \lambda_{i,2} = 2$, $\sigma_{i,1} = \sigma_{i,2} = 20$, $\tau_{i,2} = 0.005$, $l_{i,1} = diag[136.49, 192.94, 18.07]$, $l_{i,2} = diag[154.47, 240.93, 0.01]$. And we choose the initial conditions as $x_{1,1}(0) = [0.02, -0.3, -0.01]^T$, $x_{2,1}(0) =$

$x_{3,1}(0) = x_{4,1}(0) = [0.03, -0.4, 0]^T$, $x_{i,2}(0) = [0, 0, 0]^T$, $\hat{x}_{1,1}(0) = [0.02, -0.3, -0.01]^T$, $\hat{x}_{i,1}(0) = x_{i,1}(0)$, $\hat{x}_{i,2}(0) = [0, 0, 0]^T$.

Simulation results are shown in Figs. 1-4. Fig. 1 shows the leader's and UAVs' angles and their observations under the attacks above parameters. Fig. 2 shows the leader's and UAVs' angular velocities and their observations. Both figs exhibit that the followers' attitude can be consistent with the leader's under the sparse sensor attacks. The tracking errors are described in Fig. 3, from which we can see that the tracking error converges to zero asymptotically. Fig. 4 represents the control output of four UAVs.

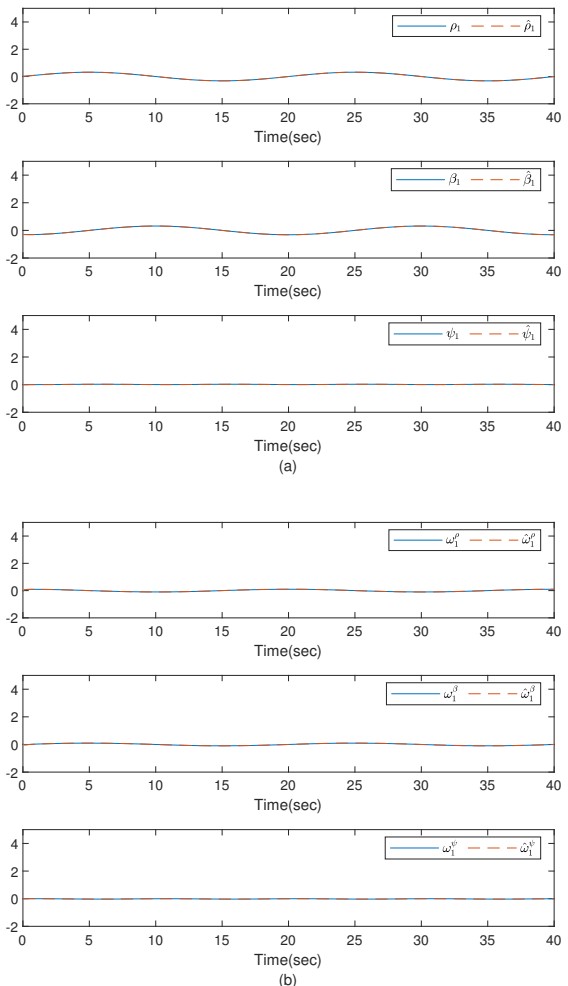

Fig. 1.   (a) Curves of UAV1's angles $x_{i,1} = [\rho_i, \beta_i, \psi_i]^T$ and UAV1's angles estimates $\hat{x}_{i,1} = [\hat{\rho}_i, \hat{\beta}_i, \hat{\psi}_i]^T$. (b) Curves of UAV1's angular velocities $x_{i,2} = [\omega_i^\rho, \omega_i^\beta, \omega_i^\psi]^T$ and UAV1's angular velocities estimates $\hat{x}_{i,2} = [\hat{\omega}_i^\rho, \hat{\omega}_i^\beta, \hat{\omega}_i^\psi]^T$.

## V. CONCLUSION

This paper studied the secure consensus tracking control problem of second-order nonlinear MASs against the sparse sensor attacks. By designing a secure data selector to extract

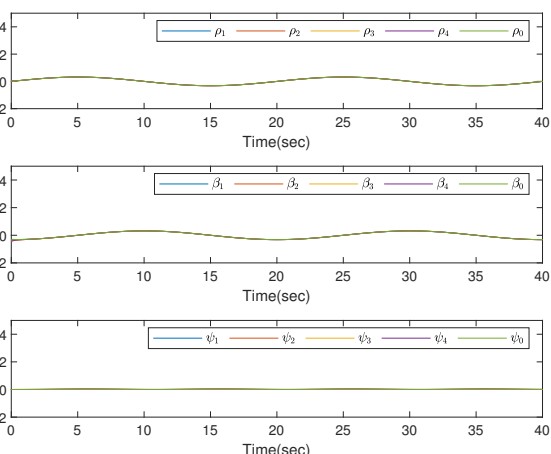

Fig. 2.   Four UAVs' angles and leader's angles.

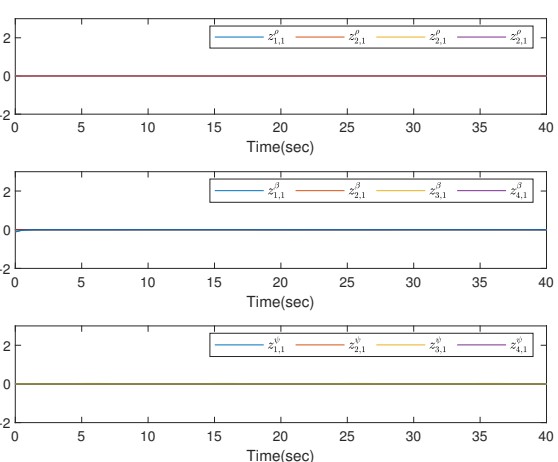

Fig. 3.   Cooperative errors $z_{i,1}$.

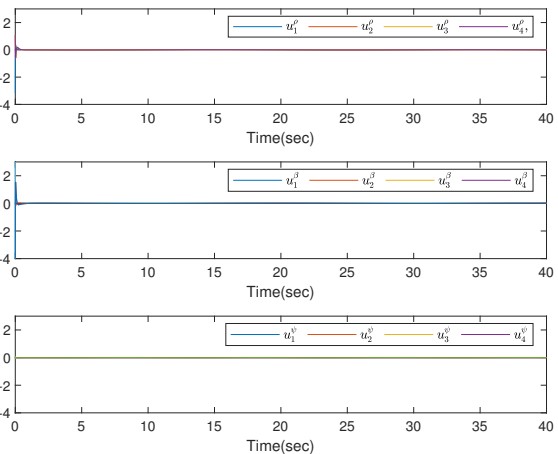

Fig. 4.   Control output of four UAVs.

a set of unattacked sensor data and using an NN secure state observer to reconstructed the unavailable system states, adaptive NN consensus controller via dynamic surface control technique is employed to ensure that all signals of the closed-loop systems are ultimately bounded. Furthermore, the consensus tracking control errors between followers and the leader converge to a small neighborhood. Finally, the effectiveness of the proposed control method is proved through a simulation of multiple UAVs.

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
