# OpenReview forum: "Adaptive NN Consensus Control for Second-Order Nonlinear Multi-Agent Systems Against Sparse Sensor Attacks"
_IEEE.org/ICIST/2024/Conference — IEEE ICIST 2024 Conference Submission_

### Official Review · Reviewer_QQVS · 2024-08-21
**Adaptive NN Consensus Control for Second-Order Nonlinear Multi-Agent Systems Against Sparse Sensor Attacks**

**Rating:** 6
**Confidence:** 4

**Review:**

1. What is the principle of data selection for the designed secure data selector when the system is subjected to sparse attacks? In the simulation process, How does the selector perform data selection in the actual simulation process?
2. The secure state observer has been proposed to estimate unknown states. How can the accuracy of the observation mechanism be ensured, and is there any theoretical analysis of observation errors?
3. In graph theory, the definition of $0$ is repetitive, such as $a_{i,j}=0$ and the leader $0$. Please clarify the definition of the parameters to ensure their uniqueness.

---

### Official Review · Reviewer_oCN4 · 2024-08-22
**This paper investigates the secure consensus tracking control problem of second-order nonlinear multi-agent systems against sparse sensor attacks. The feasibility of the designed control approach is proven via the simulation example. However, the following suggestions need to be considered in the revised manuscript to further improve the quality of this paper.**

**Rating:** 7
**Confidence:** 3

**Review:**

1. How does the proposed secure data selector effectively distinguish unattacked output data from measurements affected by sensor attacks? What criteria or methods are used to ensure the accuracy and reliability of the selected data?
2. Could the authors elaborate on the construction of the NN secure state observer? How does it leverage the unattacked output data to estimate unavailable system states?
3. What are the conditions under which the proposed method guarantees bounded tracking errors and ultimately bounded signals of the closed-loop systems?

---

### Official Review · Reviewer_GM1i · 2024-08-27
**The topic under consideration is interesting. This paper can be accepted after minor modifications.**

**Rating:** 8
**Confidence:** 3

**Review:**

This paper investigates the secure consensus tracking control problem of second-order nonlinear multi-agent systems against sparse sensor attacks. The topic under consideration is interesting. Detailed comments and suggestions are listed as follows.
1.	Some symbols for use are not clear, such as “$\bar y_i=C_i\bar x_i=[x_{i,1},\ldots,x_{i,1}]$”, “the definition of the operator $Med[\cdot]$”, and “the symbol of equation (3)” in Page 2.
2.	In Problem statement, it is mentioned that “there exist $N$ followers”. However, in the following，the virtual leader is mentioned and the actual leader is not mentioned. Is there an actual leader in the system？
3.	The English writing of the paper needs to be further polished, and some typos should be fixed.

---

### Decision · Program_Chairs · 2024-09-06

Accept (Oral)